# The prevalence of depressive disorder and its association in Thai cervical cancer patients

**Nuntaporn Karawekpanyawong**[1]*, **Kewalee Kaewkitikul**[1], **Benchalak Maneeton**[1], **Narong Maneeton**[1], **Sitthicha Siriaree**[2]

1 Faculty of Medicine, Department of Psychiatry, Chiang Mai University, Chiang Mai, Thailand, 2 Faculty of Medicine, Department of Obstetrics and Gynecology, Chiang Mai University, Chiang Mai, Thailand

* nuntaporn.karawek@cmu.ac.th

## Abstract

### Purpose

The purpose of this study is to examine the prevalence, associated factors and quality of life associated with depressive disorder in cervical cancer patients.

### Patients and methods

This cross-sectional study was carried out in a gynecologic oncology clinic of a university hospital in Northern Thailand from October 2018 to August 2019. Two-hundred cervical cancer patients were screened for depressive disorder using the nine-item Patient Health Questionnaire (PHQ-9), and psychiatrists interviewed eligible patients to confirm diagnoses. We measured the quality of life using questionnaires from the European Organisation for the Research and Treatment of Cancer: Quality of Life Questionnaire Core 30 (EORTC QLQ-C30) and Cervical Cancer Module 24 (EORTC QLQ-Cx24). Associated factors, including comorbidity, fatigue, and pain, were collected using the Charlson Comorbidity Index (CCI), the eleven-item Chalder Fatigue Scale (CFQ 11), and the visual analog scale (VAS) for pain, respectively.

### Results

Twenty-seven (13.5%) cervical cancer patients were diagnosed with depressive disorder by psychiatrists according to the DSM-5. Depressive disorder was related to a worse quality of life in these patients. A binary logistic regression analysis revealed that depressive disorder among these patients was linked with these factors: high fatigue score (aOR: 1.35; CI: 1.18–1.53), high pain score (aOR: 1.25; CI: 1.02–1.54), no perception of social support, (aOR: 3.12; CI: 1.11–8.81), and no previous surgical treatment for cervical cancer (aOR: 2.99; CI: 1.08–8.29).

### Conclusion

The depressive disorder prevalence was 13.5% in Northern Thai cervical cancer patients. In this demographic, cervical cancer patients—who reported high fatigue or pain scores, did

**Data Availability Statement:** All our anonymized data set files are available from the figshare database (DOIs: 10.6084/m9.figshare.14124017.).

**Funding:** This work was supported by a grant from the Faculty of Medicine, Chiang Mai University, Thailand to NK (grant number 043/2562), and Chiang Mai University, Thailand to BM (grant number 4/2564). The funders had no role in study design, data collection, and analysis, decision to publish, or preparation of the manuscript.

**Competing interests:** The authors have declared that no competing interests exist.

not perceive social support, or had no previous cervical cancer surgery- were more likely to have depressive disorder.

## Introduction

Cervical cancer is the fourth most common cancer among women globally [1], the third most common among Asian women [2], and the second most common cancer in Thailand [3]. Cervical cancer incidence is high in Eastern and Northern Thailand, respectively [3].

In the general population, depressive disorder is the most prevalent psychiatric disorder, at about 5.0 to 17.0% worldwide [4,5]. The prevalence of depressive disorder among Thai people is 0.3 to 2.4%, but much higher among cervical cancer patients ranging from 4.2 to 80.0% [6–17]. The wide range of depressive disorder prevalence among Thai cervical cancer patients might be explained by differences in culture and diagnostic measures. In Thailand, only two studies have examined the association between depressive disorder and overall gynecologic cancers (ovarian cancer, cervical cancer, uterine cancer, and gestational trophoblastic tumor) [17,18]; however, the association between depressive disorder and cervical cancer has not been explicitly elucidated.

There are several treatment methods to cure cervical cancer, depending on cancer stage [19]. Cervical cancer treatment is generally effective, with 5-year survival rates around 70.0%. Because of the high survival rate after treatment, a cervical cancer patients' quality of life becomes an important issue to be considered [20]. Factors which are detrimental to cervical cancer patients' quality of life include persistent gynecologic problems, low social support, somatic symptoms, less adaptive coping, medical conditions, sleep problems, inadequate education, and depression [21–24]. However, discrete protocols to monitor and improve their quality of life have not been established.

Depressive disorder in cervical cancer patients has been reported to decrease quality of life [21,22,24] and affect cancer outcomes in several ways, such as; decreased motivation to obtain healthcare [25], reduced adherence to medications [26], lengthening hospitalization [27], and increasing morbidity [28,29]. Not only can cervical cancer cause depressive disorder through its physical, emotional, and psychological effects, but the treatment may also lead to depressive disorder due to the involvement of reproductive and hormonal systems, thereby affecting the identity, self-image, and self-esteem of the patient [9,14,16,17].

Previous studies have reported a wide range of factors associated with depressive disorder among cervical cancer patients, such as age [7–9,13,14,16,17], education [7–9,13,14,16,17], marital status [7–9,13,14,16,17], socioeconomic status [7–9,13,14,16,17], pain [7,8,14,18], fatigue [7,8,14,18], menopausal symptoms [7,8,14,18], comorbidities [12,13], stage of cervical cancer [13,16,17], and type of treatment [13,16,17]. The heterogeneity of these factors makes them challenging to deal with in clinical practice. Thus, factors associated with depressive disorder among cervical cancer patients might be beneficial to explore in our local context.

Therefore, this study aimed to investigate the prevalence of depressive disorder in cervical cancer patients in Northern Thailand, examine their quality of life, and identify factors associated with depressive disorder among these patients.

## Material and methods

### Study design

This study is a cross-sectional study conducted at a gynecologic oncology clinic of a university hospital that included patients from all provinces in Northern Thailand. The demographic of

this study is aimed to represent cervical cancer patients who are receiving treatment from hospitals in Northern. The Research Ethics Committee, Faculty of Medicine, Chiang Mai University has approved this study protocol (reference number 399/5747).

## Participants

We invited all cervical cancer patients who visited the gynecologic oncology clinic from October 2018 to August 2019 to participate in this study until the intended sample size was reached (n = 200) without selection. The specific demographic of this study identified were patients diagnosed with cervical cancer aged 18 years or older who could communicate in Thai and were willing to participate in the study. We excluded patients who were unable to read/write in Thai or had history of significant cognitive impairment. After researchers thoroughly explained the details of the study, all participants gave written informed consent before their participation.

## Assessment

We collected data through questionnaires, review of medical records, and interviews. The patients also completed all questionnaires in a single visit at the gynecologic oncology clinic.

**Depressive disorder.** The participants were divided into two groups; the depressed and non-depressed this was achieved in two ways. First, we screened depressive disorder using a self-rating questionnaire, the nine-item Patient Health Questionnaire (PHQ-9; Thai version) [30–32]. This questionnaire asked the frequency of feeling relevant to the depressive symptoms in the past two weeks. Some examples of the available choices given to the patient were "Feeling down, depressed, or hopeless" and "Little interest or pleasure in doing things.". Each question scored from 0 (not at all) to 3 (nearly every day). The recommended cut-off points for possible depressive disorder was nine or greater. The questionnaire's sensitivity and specificity were obtained by comparing the screening data with the Thai version of the Mini International Neuropsychiatric Interview (MINI). The sensitivity at this cut-off value was 0.84, and the specificity was 0.77. Second, to prevent a false positive from the screening test, we implemented clinical diagnosis interviews conducted by psychiatrists using a full psychiatric assessment according to DSM-5 classification [33]. Since the somatic symptoms of depressive disorder (decreased appetite, loss of energy, insomnia, loss of sexual drive, and psychomotor retardation) were measured in the screening tool, it is difficult to differentiate with symptoms seen in cancer patients. The depressed group included participants who were diagnosed with any depressive disorder at the time the study was conducted.

We assessed suicide risk by using a suicidal screening form (the 8Q) which was developed from the suicidality module of the Mini International Neuropsychiatric Interview (M.I.N.I.) [34]. Which aim to identify suicidal ideations, suicidal plans, and suicidal attempts in the past month. Scoring scheme cut-off points used are the same as M.I.N.I. Cut points of 8, 16, and 17 might be interpreted as representing a mild, moderate, and high risk of suicidality.

**Quality of life.** We assessed the quality of life by using the European Organization for the Research and Treatment of Cancer Quality of Life Questionnaire Core 30 (EORTC QLQ-C30) and Cervical Cancer Module (EORTC QLQ-Cx24). The EORTC Quality of Life Group provided this instrument. We have also obtained permission to use for these in our study.

The EORTC QLQ-C30 assesses the activity limitations experienced in cancer patients. Its 30 items include functional difficulty and symptom scales. The functional difficulties include global health and quality of life scale, physical, role, cognitive, emotional, and social functioning. The symptom scales include fatigue, pain, nausea or vomiting, dyspnea, appetite loss, insomnia, constipation, diarrhea, and the perceived financial impact. The levels of problems and symptoms ranged from 1 (not at all) to 4 (very much). A higher score represents a better functioning level

for the functioning scales and the global quality of life scale. For the symptom scales and items, a high score corresponds to a higher level of symptomatology. Cronbach's a coefficients of the six scales were above 0.70, except for 0.50 0f cognitive and 0.63 of social function scales [35,36].

The EORTC QLQ-Cx24 was a 24-item questionnaire to assess patient's perception of symptoms related to cervical cancer. It consisted of functioning domains (body image, sexual activity, sexual enjoyment, and sexual/vaginal functioning) and symptoms domains (symptoms experienced, lymphoedema, peripheral neuropathy, menopausal symptoms, and sexual worry). The higher scores of all areas except sexual activity and sexual enjoyment refer to the low quality of life-related to cervical cancer. The reliability of the Thai-version EORTC QLQ-Cx24 using internal consistency, with Cronbach's alpha equal to 0.79 [37].

**Factors associated with depressive disorder.**   Demographic data of interest included age, educational level, marital status, number of children, perception of external support, family history of psychiatric disorder, personal income per month, financial problems, menopause, and menopausal symptoms. We used questionnaires (see S1 and S2 Tables) and reviewed the duration of cancer, type of cancer, cancer stage, treatment regimens, and treatment stage from medical records.

Sixteen physical illnesses affecting mortality rates were assessed using the Charlson Comorbidity Index (CCI) [38]. Comorbid diseases with a score of 1 included myocardial infarction, congestive heart failure, peripheral vascular disease, cerebrovascular disease, dementia, chronic obstructive pulmonary disease, connective tissue disease, peptic ulcer, uncomplicated diabetes mellitus, and mild liver disease. More severe illnesses, including diabetes mellitus with end-organ damage, moderate to severe chronic kidney disease, hemiplegia, leukemia, malignant lymphoma, and solid tumor, scored 2. Moderate to severe liver disease scored 3. Metastatic cancer and Acquired Immune Deficiency Syndrome scored 6.

We used the visual analog scale (VAS) for evaluating pain [39]. The VAS asks patients to rate their current pain intensity by scoring from 0 (no pain) to 10 (worst possible pain).

Fatigue was evaluated by the eleven-item Chalder Fatigue Scale (CFQ 11) with a Likert scoring system [40]. The 11-item chalder fatigue scale measures physical fatigue (questions 1–7) and mental fatigue (questions 8–11). Some examples of questions given to the patient were "Do you have problems with tiredness?" and "Do you have difficulty concentrating?". Participants can score 0 (less than usual) to 3 (much more than usual) in the Likert scoring system.

## Sample size calculation

Sample size was calculated using "N = $Z^2$ (PQ) / $E^2$", where Z = 1.96 (95% confidence level). P is the percentage of the study phenomena from a previous study. We used P = 0.078, derived from the prevalence of depressive disorder in cervical cancer from the most recent Asia study [11]. Q or 1-P was equal to 0.922. E refers to the percentage of accepted type 1 errors in the sample; we used E = 5.0% or 0.05. The calculated sample size was 110. According to response rate for survey of about 50.0% from the previous study [10], 200 patients were recruited to participate in the current study.

## Statistical analysis

All analyses were done using SPSS version 22.00 (SPSS Inc, Chicago, Ill) and the Statistically significant level was set at a p-value of < 0.050 (two-sided). We also used descriptive statistics and logistic regression analyses in this study.

All sociodemographic, clinical characteristics were compared between the depressed and non-depressed groups. We presented means with standard deviations or medians with interquartile ranges (depending on the distribution) for continuous data and frequencies with

percentages for categorical data. Chi-square test, independent t-test, or Mann–Whitney U test were used for comparison. Personal income was grouped into four groups according to the current minimum wage (315–325 baht per day) [41] and average living expenses in Thailand (300–499 baht per day) [42].

The number of children was grouped into three groups according to reproductive concern [43]. We calculated the prevalence of depressive disorder and risk of suicidality in percentages.

Logistic regression was used to examine relationships between depressive disorder and quality of life and evaluate contributing factors for depressive disorder in cervical cancer patients. We compared each patient's quality of life between the depressed and non-depressed groups by univariable logistic regression. Next, we investigated factors for depressive disorder among cervical cancer patients by including potential factors from univariable logistic regression (p < 0.200) and analyzing them in a multivariable logistic regression model.

## Results

### Study population

This study was carried out in the gynecologic oncology clinic of a university hospital in Northern Thailand from October 2018 to August 2019. A total of 200 female patients who had cervical cancer and visited the clinic to enroll in the study. Table 1 presents characteristics of the study population. Participants' mean age was 55.3 ± 0.7 years. Almost half of them had stage I cervical cancer, while about one-third of them were at stage II. They had relatively low socio-economic status. Sixty percent of them had primary school education level. Less than 20.0% of them had personal income more than minimum wage (300 baht per day). More than half of them reported they have financial problems. According to descriptive analysis, the depressed group had significantly worse financial problems, more advanced cancer, and treatment than the comparison group.

### Prevalence of depressive disorder among cervical cancer patients

We found that the prevalence of depressive disorder in this demographic was 13.5%. Thirty-one patients (15.5%) had positive results from PHQ-9 questionnaire; however, only twenty-seven patients (13.5%) were diagnosed with depressive disorder by psychiatrists according to DSM-5 classification. Among them, twenty-six patients had a major depressive disorder, and one patient had a persistent depressive disorder with intermittent major depressive episode without current episode. We found that thirteen patients (6.5%) had suicidal ideation, but only one patient had moderate to high suicidal risk (0.5%).

### Quality of life and depressive disorder

The depressed group had significantly lower EORTC QLQ-C30 functioning scores in all areas including global health status, physical functioning, role functioning, emotional functioning, cognitive functioning, and social functioning (Fig 1). This result represented a worse level of functioning in the depressed group.

Moreover, almost all the means of EORTC QLQ-C30 symptoms scores in the depressed group were significantly higher than the non-depressed group, including fatigue, pain, nausea or vomiting, dyspnea, appetite loss, insomnia, and the perceived financial difficulties. Whereas the means of constipation and diarrhea scores in the depressed group were higher, but they did not have statistical differences (Fig 2).

The means of symptoms experience score, body image score, and sexual worry score from EORTC QLQ-Cx24 in the depressed group were significantly higher. Likewise, the depressed

**Table 1. Characteristics of the study population.**

| Characteristics, n (%)/Mean ± SD/Median [IQR] | All (n = 200) | Depressed group (n = 27) | Non-depressed group (n = 173) | p-value |
|---|---|---|---|---|
| **Age (years)** | 55.3 ± 0.7 | 52.7 ± 1.7 | 55.8 ± 0.8 | 0.147 [c] |
| **Educational level** | | | | 0.434 [a] |
| Uneducated | 28 (14.0%) | 6 (22.2%) | 22 (12.7%) | |
| Primary school | 123 (61.5%) | 15 (55.6%) | 108 (62.4%) | |
| Secondary school | 23 (11.5%) | 4 (14.8%) | 19 (10.9%) | |
| Higher education | 26 (13.0%) | 2 (7.4%) | 24 (13.9%) | |
| **Marital status** | | | | 0.933 [a] |
| Single | 8 (4.0%) | 1 (3.7%) | 7 (4.0%) | |
| Married | 146 (73.0%) | 21 (77.8%) | 125 (72.3%) | |
| Separated or divorced | 21 (10.5%) | 2 (7.4%) | 19 (11.0%) | |
| Widowed | 25 (12.5%) | 3 (11.1%) | 22 (12.7%) | |
| **Number of children** | | | | 0.953 [a] |
| No child | 17 (8.5%) | 2 (7.4%) | 15 (8.7%) | |
| 1 child | 48 (24.0%) | 7 (25.9%) | 41 (23.7%) | |
| ≥ 2 children | 135 (67.5) | 18 (66.7%) | 117 (67.6%) | |
| **Have family history of psychiatric disorder** | 11 (5.5%) | 1 (3.7%) | 10 (5.8%) | 0.660 [a] |
| **Personal income per month** | | | | 0.280 [a] |
| < 5000 baht | 135 (67.5%) | 22 (81.5%) | 113 (65.3%) | |
| 5001–10000 baht | 27 (13.5%) | 3 (11.1%) | 24 (13.9%) | |
| 10001–15000 baht | 15 (7.5%) | 0 (0.0%) | 15 (8.7%) | |
| > 15000 baht | 23 (11.5%) | 2 (7.4%) | 21 (12.1%) | |
| **Financial problem** | | | | **0.039** [a] |
| **No** | 83 (41.5%) | 5 (18.5%) | 78 (45.1%) | |
| Low | 49 (24.5%) | 7 (25.9%) | 42 (24.3%) | |
| Moderate | 34 (17.0%) | 7 (25.9%) | 27 (15.6%) | |
| Severe | 34 (17.0%) | 8 (29.6%) | 26 (15.0%) | |
| **Duration of cancer (months)** | 22 [3–64] | 10 [2–37] | 23 [3–71] | 0.150 [b] |
| **Type of cancer** | | | | 0.508 [a] |
| Squamous cell | 145 (72.5%) | 21 (77.8%) | 124 (71.7%) | |
| Non-squamous cell | 55 (27.5%) | 6 (22.2%) | 49 (28.3%) | |
| **Stage of cancer** | | | | **0.038** [a] |
| Stage 1 | 90 (45.0%) | 6 (22.2%) | 84 (48.6%) | |
| Stage 2 | 67 (33.5%) | 13 (48.1%) | 54 (31.2%) | |
| Stage 3, 4 | 43 (21.5%) | 8 (29.7%) | 35 (20.2%) | |
| **Stage of treatment** | | | | **0.033** [a] |
| Annual check up | 108 (54.0%) | 9 (33.4%) | 99 (57.2%) | |
| First line | 70 (35.0%) | 12 (44.4%) | 58 (33.5%) | |
| Second or third line | 22 (11.0%) | 6 (22.2%) | 16 (9.3%) | |

[a] Pearson Chi-Square;

[b] Mann-Whitney u test;

[c] Independent t-test.

Numbers in bold represent significant results with p-value < 0.050 (two-sided).

group had significantly lower sexual activity. On the other hand, the rest items' mean score, including sexual/vaginal functioning, lymphoedema, peripheral neuropathy, menopausal symptoms, and sexual enjoyment, did not show the statistical difference (Fig 3).

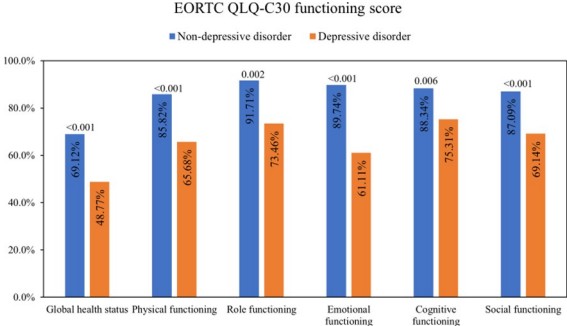

**Fig 1. The means of EORTC QLQ-C30 functioning scores of the depressed group compared to the non-depressed group, with p-values.** Representing a higher percentage of the global quality of life, functioning scales indicate a better quality of life. Abbreviations: EORTC QLQ-C30, European Organisation for Research and Treatment of Cancer Quality of Life Core Questionnaire. P-values < 0.050 were considered statistically significant.

## Factors associated with depressive disorder

From the univariable analysis, twelve factors gave p < 0.200 and were eligible to enter a multivariable logistic regression model. Univariable analysis found that the depressed group had a higher pain score, higher fatigue score and more advanced cancer stage and treatment than the non-depressed group; on the other hand, the non-depressed group had received significantly more surgical treatment for cancer than the depressed group (p < 0.050) as shown in Table 2.

After entering the multivariable model, only four factors remained significantly associated with depressive disorder in cervical cancer patients: high fatigue score (aOR: 1.35; CI: 1.18–1.53), high pain score (aOR: 1.25; CI: 1.02–1.54), no perception of social support, (aOR: 3.12; CI: 1.11–8.81), and no previous surgical treatment for cervical cancer (aOR: 2.99; CI: 1.08–8.29) as shown in Table 3.

## Discussion

### Prevalence of depressive disorder among cervical cancer patients

From our results, the prevalence of depressive disorder in cervical cancer patients in Northern Thailand is 13.5%, that is five times higher than in the Thai general population (2.4% for overall, 2.9% for female) [5], which is consistent with the previous studies from the USA and China

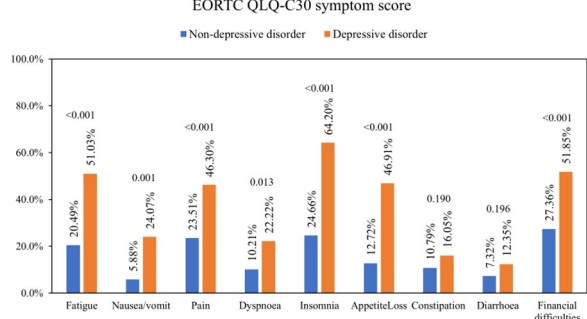

**Fig 2. The means of EORTC QLQ-C30 symptoms scores of the depressed group compared to the non-depressed group, with p-values.** A higher percentage on symptom scales indicates a poorer quality of life. Abbreviations: EORTC QLQ-C30, European Organisation for Research and Treatment of Cancer Quality of Life Core Questionnaire. P-values < 0.050 were considered statistically significant.

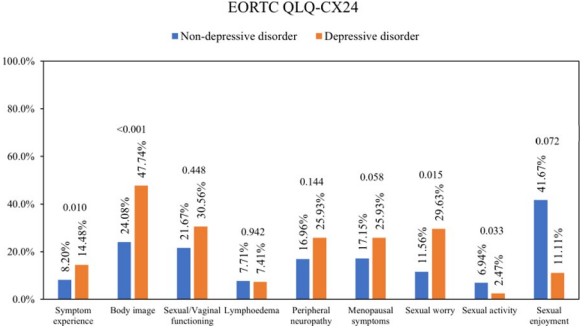

**Fig 3. The means of EORTC QLQ-Cx24 scores of the depressed group compared to the non-depressed group, with p-values.** A higher percentage on cervical-related symptom scales indicates a poorer quality of life. A higher percentage of sexual activity and sexual enjoyment indicates a better quality of life. Abbreviations: EORTC QLQ-Cx24, EORTC Cervical Cancer Module. P-values < 0.050 were considered statistically significant.

[7,13]. Depressive disorder is significantly more common among patients with cervical cancer than those with other cancers and community samples [7,44]. Among Thai patients with gynecologic cancers, patients with cervical cancer presented the second-highest depressive disorder prevalence, following patients with ovarian cancer [17].

The reasons that cervical cancer patients are more likely to have depressive disorder may lie in the consequences of cancer itself and its treatment. Cervical cancer patients, especially in advanced stage, suffer from cervical cancer symptoms, including watery foul-odor vaginal discharge, vaginal bleeding after intercourse or after menopause, chronic pelvic pain, urinary and bowel incontinence, and pain during intercourse. They also bear side effects and complications from cervical cancer treatment, including pain, nausea, vomiting, and fatigue. These symptoms can hamper patient's functional ability and affect their emotions and self-image. Moreover, the deterioration in sexual function after cervical cancer treatment brings a feeling of defeminization due to the loss of reproductive function. This distress and loss of self-esteem increases the risk of depressive disorder [16,21].

Our prevalence rate of 13.5% is lower than 21.3–80.0% of most prior studies [6,8,10,14,15,17,18], whereas this rate is higher than 4.2% and 7.8% of some studies [11,12]. We hypothesize that the difference may come from two main reasons: the difference in measurement and the difference in culture.

The lower prevalence may also be explained by two reasons: measurement and cultural context. The first explanation is that defined depressive disorder by psychiatrists' clinical interview, while prior studies that reported the higher prevalence (21.3–80.0%) used the screening tool or the structured clinical interview [6,8,10,14,15,17,18]. Their results might have included false positives due to the overlap symptoms among cancer, treatment side effects and depressive disorder, including; loss of appetite, fatigue and sleep difficulties [45]. Another explanation is that the prevalence of depressive disorder in the Thai population is generally lower than in other countries (0.3–2.4% compared to 5.0–17.0%) [4,5] this is thought to be due to Thai culture. In Thai culture it is discouraged to express emotion which increases stressor acceptance [46]. This stigmatization discourages emotional distress expression and can affect physician's diagnoses of depressive disorder [47,48]. Furthermore, many Thai people believe in superstitions like ghosts and spirits, combined with Buddhist concepts like merit and karma. These beliefs also contribute to increase stressor acceptance among Thai people [46].

In contrast, the amount of cases retrieved from our study is higher than the 4.2–7.8% of cases from studies that defined depressive disorder by using diagnosis from the ICD code

**Table 2. Univariable logistic regression for potential associated factors of depressive disorder among cervical cancer patients.**

| Characteristics, n (%) / Median [IQR] | All (n = 200) | Depressed group (n = 27) | Non-depressed group (n = 173) | OR (95% CI) | p-value |
|---|---|---|---|---|---|
| **Age (years)** | 56 [49–62] | 51 [48–58] | 57 [50–62] | 0.97 (0.93–1.01) | 0.148 |
| **Educational level** | | | | | 0.452 |
| Uneducated | 28 (14.0%) | 6 (22.2%) | 22 (12.7%) | 1.00 | |
| Primary school | 123 (61.5%) | 15 (55.6%) | 108 (62.4%) | 0.51 (0.18–1.46) | |
| Secondary school | 23 (11.5%) | 4 (14.8%) | 19 (10.9%) | 0.77 (0.19–3.15) | |
| Higher education | 26 (13.0%) | 2 (7.4%) | 24 (13.9%) | 0.31 (0.06–1.68) | |
| **Marital status** | | | | | 0.934 |
| Single | 8 (4.0%) | 1 (3.7%) | 7 (4.0%) | 1.00 | |
| Married | 146 (73.0%) | 21 (77.8%) | 125 (72.3%) | 1.18 (0.14–10.05) | |
| Separated or divorced | 21 (10.5%) | 2 (7.4%) | 19 (11.0%) | 0.74 (0.06–9.46) | |
| Widowed | 25 (12.5%) | 3 (11.1%) | 22 (12.7%) | 0.96 (0.09–10.71) | |
| **Number of children** | | | | | 0.954 |
| No child | 17 (8.5%) | 2 (7.4%) | 15 (8.7%) | 1.00 | |
| 1 child | 48 (24.0%) | 7 (25.9%) | 41 (23.7%) | 1.28 (0.24–6.86) | |
| ≥ 2 children | 135 (67.5%) | 18 (66.7%) | 117 (67.6%) | 1.15 (0.24–5.47) | |
| **Perception of external support** | 149 (74.5%) | 16 (59.3%) | 133 (76.9%) | 0.44 (0.19–1.02) | 0.055 |
| **Have family history of psychiatric disorder** | 11 (5.5%) | 1 (3.7%) | 10 (5.8%) | 0.63 (0.08–5.10) | 0.663 |
| **Personal income per month** | | | | | 0.756 |
| < 5000 baht | 135 (67.5%) | 22 (81.5%) | 113 (65.3%) | 2.04 (0.45–9.35) | |
| 5001–10000 baht | 27 (13.5%) | 3 (11.1%) | 24 (13.9%) | 1.31 (0.20–8.62) | |
| 10001–15000 baht | 15 (7.5%) | 0 (0.0%) | 15 (8.7%) | NA | |
| > 15000 baht | 23 (11.5%) | 2 (7.4%) | 21 (12.1%) | 1.00 | |
| **Financial problem** | | | | | 0.056 |
| No | 83 (41.5%) | 5 (18.5%) | 78 (45.1%) | 1.00 | |
| Low | 49 (24.5%) | 7 (25.9%) | 42 (24.3%) | 2.60 (0.78–8.70) | |
| Moderate | 34 (17.0%) | 7 (25.9%) | 27 (15.6%) | 4.04 (1.18–13.81) | |
| Severe | 34 (17.0%) | 8 (29.6%) | 26 (15.0%) | 4.80 (1.44–15.97) | |
| **Not menopause** | 173 (86.5%) | 22 (81.5%) | 151 (87.3%) | 1.56 (0.54–4.54) | 0.415 |
| **Menopausal symptoms** | 91 (45.5%) | 17 (63.0%) | 74 (42.8%) | 2.27 (0.99–5.25) | 0.054 |
| **CCI score** | 4 [3–5] | 3 [2–6] | 4 [3–5] | 1.02 (0.86–1.21) | 0.812 |
| **VAS for pain score** | 2 [0–5] | 5 [3–5] | 2 [0–5] | 1.34 (1.13–1.58) | **0.001** |
| **CFQ 11 fatigue score** | 16 [13–20] | 21 [18–23] | 16 [13–19] | 1.38 (1.21–1.56) | **< 0.001** |
| **Duration of cancer (months)** | 22 [3–64] | 10 [2–37] | 23 [3–71] | 0.99 (0.99–1.00) | 0.102 |
| **Type of cancer** | | | | | 0.510 |
| Squamous cell | 145 (72.5%) | 21 (77.8%) | 124 (71.7%) | 1.00 | |
| Non-squamous cell | 55 (27.5%) | 6 (22.2%) | 49 (28.3%) | 0.72 (0.28–1.90) | |
| **Stage of cancer** | | | | | **0.049** |
| Stage 1 | 90 (45.0%) | 6 (22.2%) | 84 (48.6%) | 1.00 | |
| Stage 2 | 67 (33.5%) | 13 (48.1%) | 54 (31.2%) | 3.37 (1.21–9.40) | |
| Stage 3, 4 | 43 (21.5%) | 8 (29.7%) | 35 (20.2%) | 3.20 (1.03–9.90) | |
| **Recurrent of cancer** | 16 (8.0%) | 2 (7.4%) | 14 (8.1%) | 0.91 (0.20–4.24) | 0.903 |
| **Metastasis of cancer** | 3 (1.5%) | 1 (3.7%) | 2 (1.2%) | 3.29 (0.29–37.57) | 0.338 |
| **Previous surgery** | 120 (60.0%) | 11 (40.7%) | 109 (63.0%) | 0.40 (0.18–0.92) | **0.032** |
| **Previous radiation therapy** | 149 (74.5%) | 22 (81.5%) | 127 (73.4%) | 1.59 (0.57–4.46) | 0.374 |
| **Previous chemotherapy** | 134 (67.0%) | 22 (81.5%) | 112 (64.7%) | 2.40 (0.86–6.65) | 0.093 |
| **Previous hormones** | 8 (4.0%) | 1 (3.7%) | 7 (4.0%) | 0.91 (0.11–7.72) | 0.933 |
| **Previous brachytherapy** | 59 (29.5%) | 9 (33.3%) | 50 (28.9%) | 1.23 (0.52–2.92) | 0.639 |

(*Continued*)

**Table 2.** (Continued)

| Characteristics, n (%) / Median [IQR] | All (n = 200) | Depressed group (n = 27) | Non-depressed group (n = 173) | OR (95% CI) | p-value |
|---|---|---|---|---|---|
| **Current chemotherapy** | 71 (35.5%) | 14 (51.9%) | 57 (32.9%) | 2.19 (0.97–4.97) | 0.060 |
| **Stage of treatment** | | | | | **0.042** |
| Annual check up | 108 (54.0%) | 9 (33.4%) | 99 (57.2%) | 1.00 | |
| First line of chemotherapy | 70 (35.0%) | 12 (44.4%) | 58 (33.5%) | 2.28 (0.90–5.73) | |
| Second or third line of chemotherapy | 22 (11.0%) | 6 (22.2%) | 16 (9.3%) | 4.13 (1.29–13.16) | |

IQR, Interquartile ranges; OR, odds ratio; CI, confidence interval; CCI, the Charlson Comorbidity Index; CFQ 11, the eleven-item Chalder Fatigue Scale; VAS, the visual analog scale

Number in bold are for significant p-value at < 0.050 (two-sided).

database [11,12]. Using the database (secondary data) can introduce a high probability of underdiagnosis [49], thus giving a lower prevalence.

## Quality of life and depressive disorder

During this study we came to the conclusion that depressive disorder was correlated with a poorer quality of life, this assessment correlates to many other studies that have also been conducted [13,21,22,24]. Depressive disorder has shown to decrease patients' physical and sexual function, aggravate their cancer symptoms and raise their anxiety levels associated to body image and pain, especially during sexual activity.

The association between poor quality of life and depressive disorder might be directly correlated. The poor quality of life due to cervical cancer may come from their deteriorated functional activity (physical and sexual activity) and cancer symptoms; which can increase chances of developing depressive disorder [16]. Likewise, a negative self-image in cervical cancer patients with depressive disorder can lead to disparagement of their physical, emotional, cognitive, and social functioning. Furthermore, anhedonia or inability to feel pleasure decreases their social interaction, ability to perform daily roles, emotional satisfaction, and interest in sexual activity. Which in turn emphasizes patients' somatic symptoms [48,50].

## Factors associated with depressive disorder

The evidence gathered reveals that factors correlated with depressive disorder in cervical cancer patients were fatigue, pain, a lack of social support and a lack of surgical cancer treatment.

**Table 3. Multivariable logistic regression for independent associated factors of depressive disorder among cervical cancer patients.**

| Characteristics | aOR (95% CI) | p-value |
|---|---|---|
| **CFQ 11 fatigue score** | 1.35 (1.18–1.53) | < 0.001 |
| **VAS for pain score** | 1.25 (1.02–1.54) | 0.031 |
| **No perception of external support** | 3.12 (1.11–8.81) | 0.031 |
| **No previous surgery** | 2.99 (1.08–8.29) | 0.035 |

aOR, adjusted odds ratio; CI, confidence interval; CCI, the Charlson Comorbidity Index; CFQ 11, the eleven-item Chalder Fatigue Scale; VAS, the visual analog scale.

Adjusted for age, perception of external support, financial problems, menopausal symptoms, pain score, fatigue score, duration of cancer, stage of cancer, previous surgery, previous chemotherapy, current chemotherapy, and stage of treatment.

The p-value < 0.050 was determining statistically significant (two-sided).

Fatigue often coincides and causes depressive disorder in biopsychosocial ways [51–53]. Fatigue shares biological dysfunctions with depressive disorder, including serotonin (5-HT) dysregulation, hypothalamic-pituitary-adrenal (HPA) axis dysfunction, circadian rhythm disruption, vagal afferent nerve activation, and cytokine dysregulation. It also triggers a sense of low self-esteem and depressive disorder due to daily activity limitations [54,55]. We discovered evidence consistent with these explanations, that fatigue was a highly significant factor associated with depressive disorder (with narrow CI). Depressive disorder and pain share biological pathways and neurotransmitters [7,8,14,18,56,57]. Positive or negative emotions can modify pain perceptions via pain modulation pathways. Dysfunctions of serotonin and norepinephrine neurotransmitters cause both depressive disorder and pain. Moreover, depressive disorder can lower pain thresholds [58]. Our results supported this explanation; though our lower bound of 95% CI was near one due to a too small sample size, a prior study confirmed this association with OR 1.49 (1.19–1.86) [18]. Social support biologically and psychologically influences vulnerabilities and resilience to stress [59,60]. Consistent with several studies [10,44,61,62], cervical cancer patients who did not perceive social support were more likely to have depressive disorder. Social support affects the hypothalamic-pituitary-adrenocortical (HPA) system, the noradrenergic system, and central oxytocin pathways [59]. It also psychologically increases an important protective factor against depressive disorder- hope [60]. Thus, encouraging social support helps prevent and alleviate depressive disorder, especially support from their spouses, family, and friends [63–66]. In our study, depressive disorder was about three times more prevalent among those who did not perceive social support. However, our results presented a wide CI interval due to a small sample size.

Cervical cancer patients feel relieved after receiving cancer surgeries. We have found that patients who did not undergo surgical cancer treatment are nearly three times more likely to develop depressive disorder. However, CI interval was wide due to a small sample size. Depressive disorder was low in patients receiving surgery because they believed that their body's primary cancer had been removed. A study showed that even after cancer treatment, fear of cancer recurrence and progression could lead to depressive disorder [67]. Another study in Thailand supported this hypothesis because it found that cervical cancer survivors treated with radical hysterectomy may have a better quality of life than patients receiving chemotherapy [68].

## Clinical implications

According to the results of our study, at least one out of every ten cervical cancer patients may have depressive disorder. Therefore, depressive disorder diagnosis based on psychiatrists' clinical diagnosis in PHQ-9 positive patients is shown to be a beneficial form of diagnoses. Since our results show that the amount of cervical cancer patients that may have depressive disorder was underestimated in regular practice but overestimated by the screening questionnaire.

Optimal depressive disorder treatment might improve cervical cancer treatment and the patient's quality of life. To do that, physicians should be able to identify and modify its contributing factors.

Fatigued patients are the most vulnerable group to have depressive disorder, therefore it would be advisable that physicians pay more attention their patients and implement protocols such as frequent depressive disorder screening. Moreover, physicians can reduce depressive disorder risk by adequate pain control, encouraging social support and giving them reassurance about the effectiveness of alternative treatments other than surgery.

## Limitations

Firstly, it is important to acknowledge that we have used a convenient sample from a university hospital's gynecologic oncology clinic in Northern Thailand. Our results can be generalized to Northern Thai patients, especially in hospital settings, but may not represent larger populations in other settings or different cultural contexts. Secondly, this study was a cross-sectional study, which cannot explain the causal relationship between these factors and depressive disorder in cervical cancer patients. Thirdly, we calculated a sample size for our primary objective which is the prevalence of depressive disorder in cervical cancer, so this sample size may be too small to explore factors associated with depressive disorder among these patients. This sample size may be the explanation of our large confident intervals of some factors found. Another factor is that we did not have a control subject to compare the results between cervical cancer patents and healthy patients to identify whether the results would be affected differently regarding depressive disorder. Therefore, the application of these results to clinical settings should be made with caution. Also, the use of self-administered questionnaires might introduce recall and reporting bias. Lastly, there was a fair possibility of type II error in the interpretation of depressive disorder. Even though we have tried to reduce false-positive results from the screening using psychiatrists' interviews, we did not take an additional evaluation for patients who had negative results from the screening. Finally, we did not conduct inter-rater validity between psychiatrists.

For a more comprehensive investigation of factors related to depressive disorder in cervical cancer patients, further studies with better designs (e.g., longitudinal design, larger sample sizes, management of type II errors, and inter-rater validity) or different demographics should be conducted. We suggest researchers address psychological aspects to prevent depressive disorder in cervical cancer patients in the future.

## Conclusion

Our study shows that the prevalence of depressive disorders in cervical cancer patients in Northern Thailand is approximately 13.5% higher than in the rest of Thailand. Among cervical cancer patients, depressive disorder often coexists with a lower quality of life. Depressive disorder is prevalent among cervical cancer patients with a perceived lack of social support, who experience more pain, fatigue, and are without previous surgical treatment for cervical cancer. These results suggest that regular screening for depressive disorder and improvement of diagnosis for patients who have these factors may decrease occurrence of depressive disorder and improve the quality of life of cervical cancer patients.

## Supporting information

**S1 Table. The original language of the developed questionnaire was used in study.** (PDF)

**S2 Table. The English version of the developed questionnaire was used in study.** (PDF)

## Acknowledgments

Many thanks to Mr.Eric Tedstrom and Ms. Ruth Leatherman for their assistance in manuscript English-language editing. We are grateful to Professor Manit Srisurapanont and Mr. Suttipong Kawilapat, Department of Psychiatry, Faculty of Medicine, Chiang Mai University, for their statistical advice.

## Author Contributions

**Conceptualization:** Nuntaporn Karawekpanyawong, Kewalee Kaewkitikul, Benchalak Maneeton, Narong Maneeton, Sitthicha Siriaree.

**Data curation:** Nuntaporn Karawekpanyawong, Kewalee Kaewkitikul.

**Formal analysis:** Nuntaporn Karawekpanyawong, Kewalee Kaewkitikul.

**Funding acquisition:** Nuntaporn Karawekpanyawong, Benchalak Maneeton.

**Investigation:** Nuntaporn Karawekpanyawong, Kewalee Kaewkitikul.

**Methodology:** Nuntaporn Karawekpanyawong, Kewalee Kaewkitikul, Benchalak Maneeton, Narong Maneeton, Sitthicha Siriaree.

**Project administration:** Nuntaporn Karawekpanyawong.

**Supervision:** Benchalak Maneeton, Narong Maneeton, Sitthicha Siriaree.

**Validation:** Nuntaporn Karawekpanyawong, Kewalee Kaewkitikul, Benchalak Maneeton, Narong Maneeton, Sitthicha Siriaree.

**Visualization:** Nuntaporn Karawekpanyawong, Kewalee Kaewkitikul, Benchalak Maneeton, Narong Maneeton, Sitthicha Siriaree.

**Writing – original draft:** Nuntaporn Karawekpanyawong, Kewalee Kaewkitikul, Benchalak Maneeton, Narong Maneeton, Sitthicha Siriaree.

**Writing – review & editing:** Nuntaporn Karawekpanyawong, Kewalee Kaewkitikul, Benchalak Maneeton, Narong Maneeton, Sitthicha Siriaree.

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
