## [Decision Letter · Decision Letter 0]

11 Dec 2020

PONE-D-20-32916

The prevalence of depression and its association in Thai cervical cancer patients

PLOS ONE

Dear Dr. Karawekpanyawong,

Thank you for submitting your manuscript to PLOS ONE. After careful consideration, we feel that it has merit but does not fully meet PLOS ONE’s publication criteria as it currently stands. Therefore, we invite you to submit a revised version of the manuscript that addresses the points raised during the review process.

We look forward to receiving your revised manuscript.

Kind regards,

Stephan Doering, M.D.

Academic Editor

PLOS ONE

3. In your Methods section, please provide additional information about the participant recruitment method and the demographic details of your participants. Please ensure you have provided sufficient details to replicate the analyses such as:

- the recruitment date range (month and year)

- a description of any inclusion/exclusion criteria that were applied to participant recruitment

- a table of relevant demographic details

- a statement as to whether your sample can be considered representative of a larger population

- a description of how participants were recruited

- descriptions of where participants were recruited and where the research took place.

5. We noticed you have some minor occurrence of overlapping text with the following previous publication, which needs to be addressed:

- https://journals.plos.org/plosone/article?id=10.1371%2Fjournal.pone.0094804

In your revision ensure you cite all your sources (including your own works), and quote or rephrase any duplicated text outside the methods section. Further consideration is dependent on these concerns being addressed.

Reviewers' comments:

Reviewer's Responses to Questions

**Comments to the Author**

1. Is the manuscript technically sound, and do the data support the conclusions?

Reviewer #1: Yes

Reviewer #2: Partly

2. Has the statistical analysis been performed appropriately and rigorously? 

Reviewer #1: I Don't Know

Reviewer #2: Yes

3. Have the authors made all data underlying the findings in their manuscript fully available?

Reviewer #1: Yes

Reviewer #2: Yes

4. Is the manuscript presented in an intelligible fashion and written in standard English?

Reviewer #1: No

Reviewer #2: No

5. Review Comments to the Author

Reviewer #1: Thank you for asking me to to review this cross-sectional study of Thai cervical cancer patients. The authors identify an overall prevalence of 13.5% of depressive disorders, links to depressive disorders and quality of life and four predictors of developing these disorders. The authors should be commended for using formal psychiatric interviews to confirm the diagnosis of major depressive disorders which is often overlooked in the literature and thought out sample sizes. However, some areas for improvement of the manuscript include:

General

1. Would just re-edit for grammar/punctuation throughout to improve clarity of the manuscript as some areas are difficult to follow. e.g. “The prevalence of depression in cervical cancer patients has been vastly researched but the results have been various, 4.21% to 80%”. This is especially true within the introduction and discussion sections

Abstract

1. There is a need to improve the clarity of the methods section rather than a pure list of measures used. E.g. unclear from methods that diagnostic interviews were conducted.

2. The conclusion sentence doesn’t quite make sense. Would revise for clarity

Introduction:

1. There is a need to further highlight the importance of the research itself by discussing the impact these depressive disorders can have on cancer outcomes themselves

2. Further discussion around the secondary aims of the project and why they are required would be useful in terms of previous literature.

Methods

1. The aims and objectives should be listed at the end of the introduction section only

2. I would recommend the use of a standardised reporting framework to structure the methods section e.g. STROBE which is suitable for cross sectional studies - https://www.strobe-statement.org/index.php?id=strobe-home

3. Well done for using formal interviews for diagnosis of depression, however why was there a screening conducted prior to conductance of the interviews rather than interviews across the participants? Whilst there is good correlation between PHQ-9 and disorders is there not a risk of missing some patients screening negatively on PHQ-9?

4. How was the significance between depressed vs. non-depressed means for Quality of life outcomes measured statistically (As per figure 1-3) which is not discussed in this section?

5. What was the significance threshold for other outcomes (was this <0.05?)

Results

1. Would recommend altering subheading of ‘Comparison of quality of life between the depression and the nondepression group’ group to something a bit shorter for improved readability.

2. You mention that QofL outcomes were worse for ‘almost all of the outcomes’ for depression groups, however this isn’t the case for quite a few outcomes when evaluating the figures which demonstrate that some are not statistically significant. I would highlight which of these are not through the text to reflect this.

3. Why was the PHQ score utilised for the evaluation of the multivariable analysis rather than the more accurate diagnostic interviews from which the other prevalence figures are derived ?

Discussion

1. It would be good to compare your figures additionally to general population estimates or other cancers to offer more comparability of the psychological impact of cervical cancer specifically

2. I would add a reference for the limitations of other diagnostic methods such as the use of ICD diagnostic codes, which as you say are known to be limited by the data being inputed into the database itself.

3. Some discussion surrounding the possible mechanism around the reasons for reduced quality of life would be beneficial.

4. There is a lack of discussion surrounding the true research and clinical implications of the findings beyond ‘regular screening for depressive disorder in those patients is beneficial’ I think discussing how the identification of your findings and predictive factors identified impacts future research or clinical practice is needed e.g. through more targeted screening methods, highlighting the need to use diagnostic interviews within research etc.

Reviewer #2: This study looks at the rate of clinician-diagnosed depression using DSM 5 criteria in an opportunistic sample of women with cervical cancer of all stages presenting to a cancer clinic in northern Thailand. The aims are to document the rate of depression in this cohort, measure quality of life in depressed and non-depressed women and then explore factors associated with depression.

This study contributes to our understanding of depression in people with comorbid cancer, and extends our understanding of depression in cervical cancer. This is important as the subject field is dominated by studies of people with breast cancer.

This paper can be published but needs some revision:

1. See points added as comments in the uploaded file.

2. A thorough edit for English usage and sentence construction.

3. I would suggest removing the term 'risk factor' from the paper as this study isn't designed to determine risk factors nor attribute causality between the factors measured and the prevalence of depression. Associated factors or correlations are better phrases to explain the kinds of relationships documented here.

4. Some more detail is needed on how patients were diagnosed by the psychiatrists - interview methods for example. (Comments also in attachment). It is also of interest, but not exactly relevant to this study, to know what happened to the people diagnosed with depression. Were they offered treatment? Are they part of a cohort in another study?

5. Interpreting the rate of depression in this group (13.5%) should be flushed out at two levels. The first is comparing this data to comparable studies of other patients with cervical cancer. Once different diagnostic methods are ruled out it would be interesting to reflect on whether the rate of depression in northern Thai women with cervical cancer is different or similar to other populations with this cancer. The second context to interpret this value is within a general Thai context. The authors state that the rate of depression across the whole population is dramatically lower compared to other places - why is this? A mention of this would help interpret how much higher, relatively, the rate of depression is in Thai people with cervical cancer compared to Thai people generally. This would help reassure the reader about the scale of the impact of cervical cancer on the rate of depression.

6. PLOS authors have the option to publish the peer review history of their article (what does this mean?). If published, this will include your full peer review and any attached files.

Reviewer #1: No

Reviewer #2: No

---

## [Author Response · Author response to Decision Letter 0]

28 Apr 2021

Response to reviewer

 We would like to thank you for your careful consideration of our manuscript, your pertinent questions, and helpful suggestions. Your comments have led to several important changes to our manuscript that have considerably improved it in our opinion. We have revised our manuscript according to your recommendation, as follows:

The comments by the reviewers appear in the boxes below. The response to these comments and the associated change to the manuscript are listed underneath. The changes are also highlight in the revised manuscript.

Answer:

We have revised and made sure that our manuscript followed the PLOS ONE's style requirements.

Answer: 

I have provided the developed questionnaire in this study as supportive information (S1 and S2 Table.pdf). 

3. In your Methods section, please provide additional information about the participant recruitment method and the demographic details of your participants. Please ensure you have provided sufficient details to replicate the analyses such as:

- the recruitment date range (month and year)

- a description of any inclusion/exclusion criteria that were applied to participant recruitment

- a table of relevant demographic details

- a statement as to whether your sample can be considered representative of a larger population

- a description of how participants were recruited

- descriptions of where participants were recruited and where the research took place.

Answer:

- We have identified recruitment date in abstract and method section. Please see page 2, line 28-29 and page 5, line 98.

- We have described our inclusion and exclusion criteria in method section. Please see pages 5, lines 100-102.

- We have shown a table of participants’ characteristics in the results section. Please see pages 10, lines 218.

- We have discussed a caution about generalizability in a large population in limitations. Please see pages 21, lines 402-405.

- We have explained recruitment method; we invited all cervical cancer patients who visited the clinic at study time, informed details of the study, and asked participants for their consent. Please see pages 5, lines 97-99, 102-104.

- We have recruited all participants, assessed all questionnaires, and diagnosed depressive disorder by psychiatrists at a gynecologic oncology clinic of a university hospital. Please see pages 4, lines 91-92.

Answer:

We have uploaded our minimal anonymized data set, see DOIs: 10.6084/m9.figshare.14124017. 

5. We noticed you have some minor occurrence of overlapping text with the following previous publication, which needs to be addressed:

- https://journals.plos.org/plosone/article?id=10.1371%2Fjournal.pone.0094804

In your revision ensure you cite all your sources (including your own works), and quote or rephrase any duplicated text outside the methods section. Further consideration is dependent on these concerns being addressed.

Answer: 

We apologize for the mistake: we have modified the sentence in our manuscript to avoid plagiarism. 

Reviewer #1: 

General

1. Would just re-edit for grammar/punctuation throughout to improve clarity of the manuscript as some areas are difficult to follow. e.g. “The prevalence of depression in cervical cancer patients has been vastly researched but the results have been various, 4.21% to 80%”. This is especially true within the introduction and discussion sections

Answer: 

We have thoroughly edited our English usage to improve the clarity of our manuscript.

Abstract

1. There is a need to improve the clarity of the methods section rather than a pure list of measures used. E.g. unclear from methods that diagnostic interviews were conducted.

2. The conclusion sentence doesn’t quite make sense. Would revise for clarity

Answer: 

- We have clarified the diagnostic method and all interesting factors in the methods part. Please see pages 2, lines 29-36.

- We have changed some words to improve the clarity of the conclusion. Please see pages 2-3, lines 44-47.

 

Introduction:

1. There is a need to further highlight the importance of the research itself by discussing the impact these depressive disorders can have on cancer outcomes themselves

2. Further discussion around the secondary aims of the project and why they are required would be useful in terms of previous literature.

Answer:

-We have highlighted the impact of depressive disorder on cancer outcome. Please see page 4 line 72-74.

-We have identified the factors associated with depressive disorder, our secondary outcome, using literature review while discussing the gap of knowledge. Please see pages 4, lines 78-85.

Methods

1. The aims and objectives should be listed at the end of the introduction section only

Answer:

We have corrected this issue: we explained the objective of our study at the end of the introduction section. Please see pages 4, lines 86-88.

2. I would recommend the use of a standardised reporting framework to structure the methods section e.g. STROBE which is suitable for cross sectional studies - https://www.strobe-statement.org/index.php?id=strobe-home

Answer:

We have revised our manuscript according to a STROBE statement framework; for detail, please see below.

STROBE Statement—Checklist of items that should be included in reports of cross-sectional studies

Title and abstract 

1.(a) Indicate the study’s design with a commonly used term in the title or the abstract

Please see: Page/ Line 2/27

1.(b) Provide in the abstract an informative and balanced summary of what was done and what was found 

Please see: Page/ Line 2-3/25-47

Introduction 

Background/rationale 

2.Explain the scientific background and rationale for the investigation being reported 

Please see: Page/ Line 3-4/51-85

Objectives

3.State specific objectives, including any prespecified hypotheses 

Please see: Page/ Line 4/86-89

Methods 

Study design 

4.Present key elements of study design early in the paper 

Please see: Page/ Line 4/91

Setting

5.Describe the setting, locations, and relevant dates, including periods of recruitment, exposure, follow-up, and data collection

Please see: Page/ Line 4/91-92, 5/98

Participants

6.Give the eligibility criteria, and the sources and methods of selection of participants 

Please see: Page/ Line 5/98-102

Variables

7.Clearly define all outcomes, exposures, predictors, potential confounders, and effect modifiers. Give diagnostic criteria, if applicable 

Please see: Page/ Line 6-8/109-176

Data sources/ measurement

8.For each variable of interest, give sources of data and details of methods of assessment (measurement). Describe comparability of assessment methods if there is more than one group 

Please see: Page/ Line 6-8/109-176

Bias

9.Describe any efforts to address potential sources of bias 

Please see: Page/ Line 5/97-99, 5/106

Study size 

10.Explain how the study size was arrived at 

Please see: Page/ Line 8/177-184

Quantitative variables

11.Explain how quantitative variables were handled in the analyses. If applicable, describe which groupings were chosen and why 

Please see: Page/ Line 8-9/190-198

Statistical methods 

12.(a) Describe all statistical methods, including those used to control for confounding 

Please see: Page/ Line 8-9/185-204

12.(b) Describe any methods used to examine subgroups and interactions 

Please see: Page/ Line 8-9/185-204

12.(c) Explain how missing data were addressed 

Answer: We did not have missing data.

12.(d) If applicable, describe analytical methods taking account of sampling strategy 

Answer: We used convenience sampling.

12.(e) Describe any sensitivity analyses 

Answer: We did not have any sensitivity analysis.

Results 

Participants 

13.(a) Report numbers of individuals at each stage of study—eg numbers potentially eligible, examined for eligibility, confirmed eligible, included in the study, completing follow-up, and analysed 

Please see: Page/ Line 9/208

13.(b) Give reasons for non-participation at each stage

Answer: All invited patients were willing to participate.

13.(c) Consider use of a flow diagram

Answer: We did not have a flow diagram because we did not exclude any participants.

Descriptive data

14.(a) Give characteristics of study participants (eg demographic, clinical, social) and information on exposures and potential confounders 

Please see: Page/ Line 10/218

14.(b) Indicate number of participants with missing data for each variable of interest

Answer: We did not have missing data. 

Outcome data

15.Report numbers of outcome events or summary measures 

Please see: Page/ Line 11/222-229

Main results 

16.(a) Give unadjusted estimates and, if applicable, confounder-adjusted estimates and their precision (eg, 95% confidence interval). Make clear which confounders were adjusted for and why they were included 

Please see: Page/ Line 16/287

16.(b) Report category boundaries when continuous variables were categorized 

Please see: Page/ Line 10/218, 13/274

16.(c) If relevant, consider translating estimates of relative risk into absolute risk for a meaningful time period 

Answer: We did not use relative risk. We used the odds ratio.

Other analyses

17. Report other analyses done—eg analyses of subgroups and interactions, and sensitivity analyses

Answer: We did not conduct other analyses in this study.

Discussion 

Key results 

18.Summarise key results with reference to study objectives 

Please see: Page/ Line 16/298, 19/351-353

Limitations

19.Discuss limitations of the study, taking into account sources of potential bias or imprecision. Discuss both direction and magnitude of any potential bias 

Please see: Page/ Line 21/402

Interpretation

20.Give a cautious overall interpretation of results considering objectives, limitations, multiplicity of analyses, results from similar studies, and other relevant evidence 

Please see: Page/ Line 21/402

Generalisability

21.Discuss the generalisability (external validity) of the study results 

Please see: Page/ Line 21/402-405

Other information 

Funding

22.Give the source of funding and the role of the funders for the present study and, if applicable, for the original study on which the present article is based 

Answer: We reported financial disclosure in the journal’s submission form.

  

3. Well done for using formal interviews for diagnosis of depression, however why was there a screening conducted prior to conductance of the interviews rather than interviews across the participants? Whilst there is good correlation between PHQ-9 and disorders is there not a risk of missing some patients screening negatively on PHQ-9?

Answer: 

Due to our time limitations and lack of human resources, we could not conduct the clinical diagnostic interview across all participants. False negatives were one of the limitations in the design of this study that we have included in the limitation. Please see pages 23, lines 415-418.

4. How was the significance between depressed vs. non-depressed means for Quality of life outcomes measured statistically (As per figure 1-3) which is not discussed in this section?

5. What was the significance threshold for other outcomes (was this <0.05?)

Answer:

We have revised these issues in our manuscript to clarity the significant differences in quality of life and other variables. The significance between depressed vs. non-depressed determines the quality of life and the other variables were p<0.05. Please see pages 8, lines 186-188.

 Results

1. Would recommend altering subheading of ‘Comparison of quality of life between the depression and the nondepression group’ to something a bit shorter for improved readability.

Answer:

We have changed the subheading to “Quality of life and depression”. Please see pages 11, lines 230.

2. You mention that QofL outcomes were worse for ‘almost all of the outcomes’ for depression groups, however this isn’t the case for quite a few outcomes when evaluating the figures which demonstrate that some are not statistically significant. I would highlight which of these are not through the text to reflect this.

Answer:

We have described the non-significant result as suggested. Please see pages 12-13, lines 242-258.

3. Why was the PHQ score utilised for the evaluation of the multivariable analysis rather than the more accurate diagnostic interviews from which the other prevalence figures are derived ?

Answer:

We apologized for our mistake in the table. We analyzed using the clinical diagnosis interview to classify depressed and non-depressed groups for all analyses in our study including the multivariable analysis. We have corrected that mistake. Please see pages 16, lines 287.

Discussion

1. It would be good to compare your figures additionally to general population estimates or other cancers to offer more comparability of the psychological impact of cervical cancer specifically

Answer:

We have reviewed the prevalence of depression in cervical cancer compared with other cancer according to your suggestion. Please see pages 17, lines 301-304.

2. I would add a reference for the limitations of other diagnostic methods such as the use of ICD diagnostic codes, which as you say are known to be limited by the data being inputed into the database itself.

Answer:

We have added the reference that showed evidence of the possibility of underdiagnosis. Please see pages 18, lines 335.

3. Some discussion surrounding the possible mechanism around the reasons for reduced quality of life would be beneficial.

Answer:

We have discussed the hypotheses explaining the association between quality of life and depression in our discussion. Please see pages 18-19, lines 337-349.

4. There is a lack of discussion surrounding the true research and clinical implications of the findings beyond ‘regular screening for depressive disorder in those patients is beneficial’ I think discussing how the identification of your findings and predictive factors identified impacts future research or clinical practice is needed e.g. through more targeted screening methods, highlighting the need to use diagnostic interviews within research etc.

Answer:

-We discuss our idea in clinical implications. Please see pages 20, lines 387-400.

Reviewer #2:

1. See points added as comments in the uploaded file.

Answer: 

Abstract

● We have removed the quotation mark in conclusion. Please see pages 2-3, lines 45-47.

Introduction

● We have changed the sentence to " Factors which are detrimental to cervical cancer patients’ quality of life” to emphasizes the negative impact of depression has on quality of life. Please see pages 3, lines 66.

● We have added the references that convey the higher prevalence of depression in cervical cancer patients compared with the general population. We have revised the sentence “The prevalence of depression in cervical cancer patients has been vastly researched but the results have been various, 4.2 to 80.0%” to improve clarity. Please see pages 3, lines 55-57.

● We have explained the type of gynecologic cancer and have revised this paragraph to emphasize the lack of studies associated with depression in cervical cancer in Thailand. Please see pages 3, lines 60-61.

Method

Participants

● We have defined the period of our study. Please see pages 5, lines 98.

● We defined cognitive impairment by history taking. Please see pages 5, lines 102.

Assessment

Medical comorbidity

● Patients who reported dementia from CCI might be excluded from our studies.

Depression

● We have specified two groups as “the depressed and non-depressed groups”. Please see pages 5, lines 110-111.

● We have explained the method that we used to define the depressed group clearer. Please see pages 5-6, lines 110-126.

● The sensitivity and specificity of the PHQ-9; Thai version obtained by comparing the screening data with the Thai version of the Mini International Neuropsychiatric Interview (MINI) which is a tool served as a gold-standards for diagnosing depression. Please see pages 6, lines 117-120.

● Psychiatrists diagnosed depression by the clinical diagnosis interview with full psychiatric assessment according to DSM-5 classification. Please see pages 6, lines 120-122. 

● We changed the term “risk factor” to “associated factors” throughout our study to reduce the risk of misleading. 

Sample sized

● We increased the sample size from 110 to 200 participants due to the response rate for survey from the previous study. We have added a reference that conferred this response rate in our method. Please see pages 8, lines 183.

Results

Prevalence

● We have corrected the word “suicidal idea” with “suicidal ideation”. Please see pages 11, lines 228.

Discussion

● We have added references as you recommended.

2. A thorough edit for English usage and sentence construction.

Answer:

We have edited our English usage throughout our manuscript to improve its understandability.

3. I would suggest removing the term 'risk factor' from the paper as this study isn't designed to determine risk factors nor attribute causality between the factors measured and the prevalence of depression. Associated factors or correlations are better phrases to explain the kinds of relationships documented here.

Answer:

We changed the term “risk factor” to “associated factors” throughout our study to reduce the risk of misleading. 

4. Some more detail is needed on how patients were diagnosed by the psychiatrists - interview methods for example. (Comments also in attachment). It is also of interest, but not exactly relevant to this study, to know what happened to the people diagnosed with depression. Were they offered treatment? Are they part of a cohort in another study?

Answer:

We have given the diagnostic detail that psychiatrists diagnosed depression using the clinical diagnosis interview with full psychiatric assessment according to DSM-5 classification. Please see pages 6, lines 121-122.

We offered usual psychiatric treatment to participants diagnosed with depression and who need other psychological care such as anxiety.

This study was not part of any cohort study.

5. Interpreting the rate of depression in this group (13.5%) should be flushed out at two levels. The first is comparing this data to comparable studies of other patients with cervical cancer. Once different diagnostic methods are ruled out it would be interesting to reflect on whether the rate of depression in northern Thai women with cervical cancer is different or similar to other populations with this cancer. The second context to interpret this value is within a general Thai context. 

The authors state that the rate of depression across the whole population is dramatically lower compared to other places - why is this? A mention of this would help interpret how much higher, relatively, the rate of depression is in Thai people with cervical cancer compared to Thai people generally. 

This would help reassure the reader about the scale of the impact of cervical cancer on the rate of depression.

Answer:

We have discussed the prevalence of depression in Northern Thai cervical cancer patients in more detail using the following methods

- We have compared the prevalence of depression in cervical cancer patients and the general population or other cancers. We have discussed its hypotheses in pages 16-17, lines 298-314.

- We have compared the prevalence of depression in Northern Thai cervical cancer patients and other populations with this cancer and have discussed two possible hypothesis causes, , culture and measurement, in pages 17-18, lines 315-335. We have already given a cultural explanation for the dramatic low rate of depression in Thailand in our discussion.

---

## [Decision Letter · Decision Letter 1]

24 May 2021

The prevalence of depressive disorder and its association in Thai cervical cancer patients

PONE-D-20-32916R1

Dear Dr. Karawekpanyawong,

We’re pleased to inform you that your manuscript has been judged scientifically suitable for publication and will be formally accepted for publication once it meets all outstanding technical requirements.

Kind regards,

Stephan Doering, M.D.

Academic Editor

PLOS ONE

Reviewers' comments:

Reviewer's Responses to Questions

**Comments to the Author**

1. If the authors have adequately addressed your comments raised in a previous round of review and you feel that this manuscript is now acceptable for publication, you may indicate that here to bypass the “Comments to the Author” section, enter your conflict of interest statement in the “Confidential to Editor” section, and submit your "Accept" recommendation.

Reviewer #1: All comments have been addressed

Reviewer #2: All comments have been addressed

2. Is the manuscript technically sound, and do the data support the conclusions?

Reviewer #1: Yes

Reviewer #2: Yes

3. Has the statistical analysis been performed appropriately and rigorously? 

Reviewer #1: I Don't Know

Reviewer #2: I Don't Know

4. Have the authors made all data underlying the findings in their manuscript fully available?

Reviewer #1: Yes

Reviewer #2: Yes

5. Is the manuscript presented in an intelligible fashion and written in standard English?

Reviewer #1: Yes

Reviewer #2: Yes

6. Review Comments to the Author

Reviewer #1: Thank you for asking me to review this revised manuscript. The authors present a much improved version of the manuscript having addressed the reviewers comments satisfactorily. The methods are clearer and the findings have been put into better context within the literature. The writing has improved significantly, although a further read through to further improve some areas, particularly in the discussion would still be of benefit.

Reviewer #2: The reviewers have addressed my comments in the first round. Having looked at the editor's comments and those of reviewer 1, if these are also addressed this is a valuable paper.

7. PLOS authors have the option to publish the peer review history of their article (what does this mean?). If published, this will include your full peer review and any attached files.

Reviewer #1: No

Reviewer #2: No

---

## [Editor Report · Acceptance letter]

11 Jun 2021

PONE-D-20-32916R1 

The prevalence of depressive disorder and its association in Thai cervical cancer patients 

Dear Dr. Karawekpanyawong:

I'm pleased to inform you that your manuscript has been deemed suitable for publication in PLOS ONE. Congratulations! Your manuscript is now with our production department. 

Kind regards, 

on behalf of

Professor Stephan Doering 

Academic Editor

PLOS ONE